# The Effect of the Degree of Freedom and Weight of the Hand Exoskeleton on Joint Mobility Function

Ilham Priadythama [1,2,*], Wen Liang Yeoh [3], Ping Yeap Loh [4] and Satoshi Muraki [4]

1   Human Science International Course, Graduate School of Design, Kyushu University, Fukuoka 815-8540, Japan
2   Faculty of Engineering, Universitas Sebelas Maret, Surakarta 57126, Indonesia
3   Faculty of Science and Engineering, Saga University, Saga 840-8502, Japan; wlyeoh@cc.saga-u.ac.jp
4   Faculty of Design, Kyushu University, Fukuoka 815-8540, Japan; py-loh@design.kyushu-u.ac.jp (P.Y.L.); muraki@design.kyushu-u.ac.jp (S.M.)
*   Correspondence: priadythama@staff.uns.ac.id

**Abstract:** This study aims to investigate the effects of the degree of freedom (DOF) and weight of the hand exoskeleton (HE) on hand joint mobility function (ease of movement, movement range) in fine hand use activities. A three-digit passive HE prototype was built to fit each of the 12 participants. Two DOF setups (three DOF, two DOF), two digits' weight levels (70 g, 140 g), and barehand conditions were tested. A productivity task (performed with Standardized-Nine Hole Peg Test) and motion tasks, both performing the tip pinch and tripod pinch, were conducted to measure the task completion time and the range of motion (ROM) of the digit joints, respectively, using a motion capture system. The perceived ease rating was also measured. The results showed that DOF reduction and weight addition caused a significant task completion time increase and rating drop ($p < 0.05$). Meanwhile, the DOF reduction increased the ROM reduction of the proximal interphalangeal joints; however, the weight addition caused a correction of the ROM reduction of several joints ($p < 0.05$) at the tripod pinch. In conclusion, wearing an HE reduces hand joint mobility, especially in lower DOF. However, a certain weight addition may improve joint mobility in terms of the fingers' movement range.

**Keywords:** hand exoskeleton; joint mobility; fine hand use; nine hole peg test; motion analysis

## 1. Introduction

Disease, trauma, or aging can lead to hand function deficits [1], which may affect work productivity [2] and independence in performing activities of daily living [1]. A promising solution for improving hand function through assistive technology (AT) is the hand exoskeleton (HE) [3]. On a laboratory scale, it has been proven to improve grip strength [4,5] and partly restores some handgrip capabilities in users with disabilities [6]. However, since the first assistive purposed HE was developed [7], there remain substantial challenges in developing an HE that is practical for real-life use [8–10].

As a type of AT, the HE should effectively increase user participation in daily and occupational activities [11]. Hand activities require mobility of the hand joints, including the movement range (flexibility) and ease of movement [12], in addition to grip strength. This is supported by a previous study that revealed that approximately 50% of hand activities are fine hand uses, carried out with a precision grip [13]. However, most of the current HE designs are focused on improving the grip strength, often at the expense of reducing joint mobility in the user's hand, which eventually lowers its dexterity. Because providing both sides of these functions through HE's control system remains difficult [14], it is important to explore strategies for managing this problem.

Managing the mechanical system of the HE is a possible approach to a well-performing HE [15]. The mechanical system of an HE cannot be separated from the performance of its mechanical components, including the actuator, power transmission, and structure which

create certain mechanical characteristics of the device. As we will demonstrate in the next two paragraphs, mechanical design characteristics, such as the degree of freedom (DOF) present on the digits (fingers and thumb) and the weight of the digits, could potentially influence fine hand activities while wearing an HE.

The most common strategy to reduce the complexity of the control system of an HE is reducing the DOF by adopting an intra-finger under-actuation concept [16]. A linkage system that transmits power from a single actuator to multiple finger segments is commonly used. However, this concept has a drawback in that the movement of one segment will depend on other segments. This means that some movements will be restricted, which can lead to a reduction in hand mobility function [17–19]. However, motion restriction due to the lower DOF and the linkage friction could be beneficial because undesired motion may be suppressed which may improve precision [20].

A higher hand weight can decrease the ability to perform fine hand use activities [17]. In general, the more powerful the HE, the heavier it is. A high assistive force commonly produced by a large actuator while transmitting the force requires a thicker structure and stronger material. All these components increase the weight of the HE. As an illustration, a 1.2 kg HE can provide a 30 N grip force [21] while a 100 g system, only produces less than 2 N [22]. Even though the addition of weight may seem detrimental, counterintuitively, heavier weight may improve movement control. A case example shows that adding weight to the hand of people with impaired muscle tone can effectively suppress vibrations, thereby increasing the precision of their hand movements [23].

Previous works regarding HE development mostly focused on technological engineering, and examinations were only conducted on a narrow range of usage activities [16]. While these works are indispensable, user interaction studies on usage accommodating a wider range of activities are essential for developing HEs at higher readiness levels. Therefore, this study aims to investigate the effect of weight and DOF of the HE on hand function while performing fine hand use activities. Additionally, this study will focus on the joint mobility function to shed light on the causes for the changes while performing the activities.

This investigation requires the design and realization of an HE prototype capable of DOF and weight changes. Several participants would use this prototype to perform productivity and motion tasks that measure the task completion time and the digits' motion range, respectively. In addition, the perceived ease of performing the tasks would be rated. The results would represent the hand joint mobility (the ease of movement and the movement range) which later would be analyzed and discussed. Based on the findings we would obtain, a solution to the motion barriers in wearing the HE would be proposed.

## 2. Materials and Methods

In order to achieve the objectives of this study, and in particular, to obtain the three parameters, an experiment involving several participants was conducted. A series of steps was needed to prepare and execute the experiment, as well as analyze the results. Illustrations of these steps are systematically presented in Figure 1.

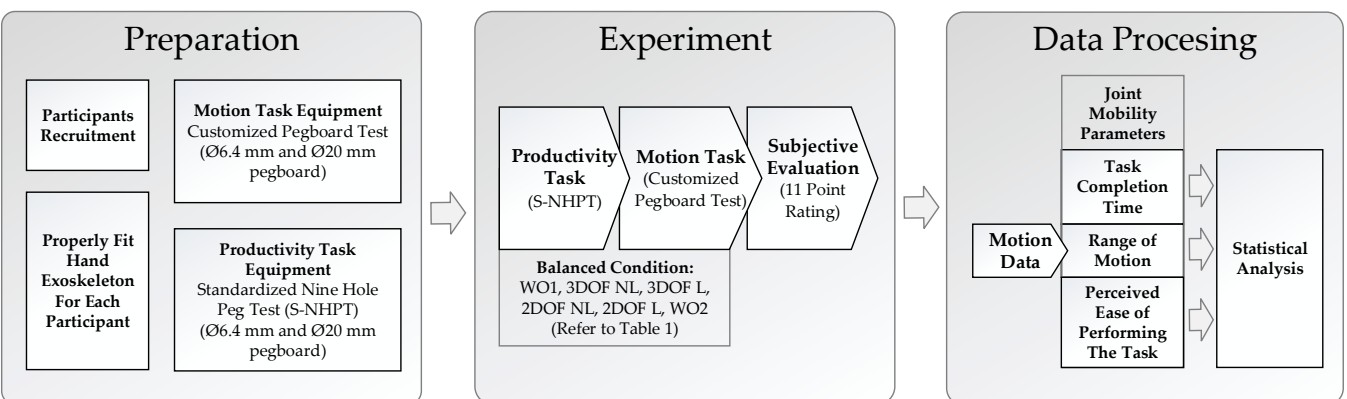

**Figure 1.** Flowchart of the experimental steps from the preparation to statistical analysis of the experiment data.

### 2.1. Participants

Twelve males, with a mean age of 26.8 ± 3.8 years, were recruited as experiment participants. All participants were right-hand dominant, tested by The Edinburgh Handedness Inventory [24]. The non-dominant hand (187.0 ± 16.3 mm hand length) was confirmed to be free of health issues and fatigue from recent strenuous hand activity. All participants provided written informed consent for the experiment, and the experiment has been approved by the Research Ethics Committee of the Faculty of Design, Kyushu University (approval number 329).

### 2.2. Prototypes

Building a proper HE prototype was the most challenging part of this study. Literature reviews have been conducted on prototypes developed in academia [3,8,16,25,26] and those developed by companies, such as Exo Hand (Festo SE & Co. KG, Esslingen, Germany). Most assistive (non-rehabilitative) prototypes adopt designs that include a rigid structure (for firm attachment [27]), three-segmented fingers (proximal-medial-distal phalanges) with a single centered joint (to connect segments), and a bar linkage mechanism for transmission. These same concepts were used in the HE prototype for this study, where the bar linkage mechanism was adapted from the Exo Hand (Festo SE & Co. KG, Esslingen, Germany). The 3-dimensional design of the prototype was built using Autodesk Fusion 360 (Autodesk Inc., San Rafael, CA, USA), as presented in Figure 2a.

The HE prototype has 26 to 28 DOFs and is equipped with three exoskeleton digits (thumb-index finger-middle finger), which is sufficient for testing the effects of using a HE on several types of precision grips. The fingertip areas were designed to be free of any HE attachments and straps to maintain fingertip sensation during pinching. To achieve this, the double-sided tape was used to attach the prototype to the participant's hand. The structure of the digits was made to be thin to minimize the contact between the index and middle finger during the tripod pinch and the distal segments were not fully covered (only 75% of distal length) by the exoskeleton to avoid contact with the ground while performing the picking up motion.

The prototype allowed for the change in DOF by installing or uninstalling two medial pushrods on the HE (Figure 2b). A medial pushrod connects the proximal pushrod to the medial segment of the exoskeleton finger. As a result, when the finger is flexing or extending, the proximal and medial segments would move simultaneously, constraining the exoskeleton finger to only two DOF. Conversely, when the connection is removed, the exoskeleton finger becomes three DOF. This DOF-changing mechanism (interdependent proximal and medial segments) was selected to avoid bulky structures near the distal segment, which can disrupt pinching. To support the motion of this linkage mechanism, a passive linear actuator with a soft spring inside (0.1 N/mm rate) was installed to provide force at the beginning of digit flexion against friction to avoid jamming. The actuator's

force at the pinch position is very small and can be neglected. Figure 2c shows how this linkage system works.

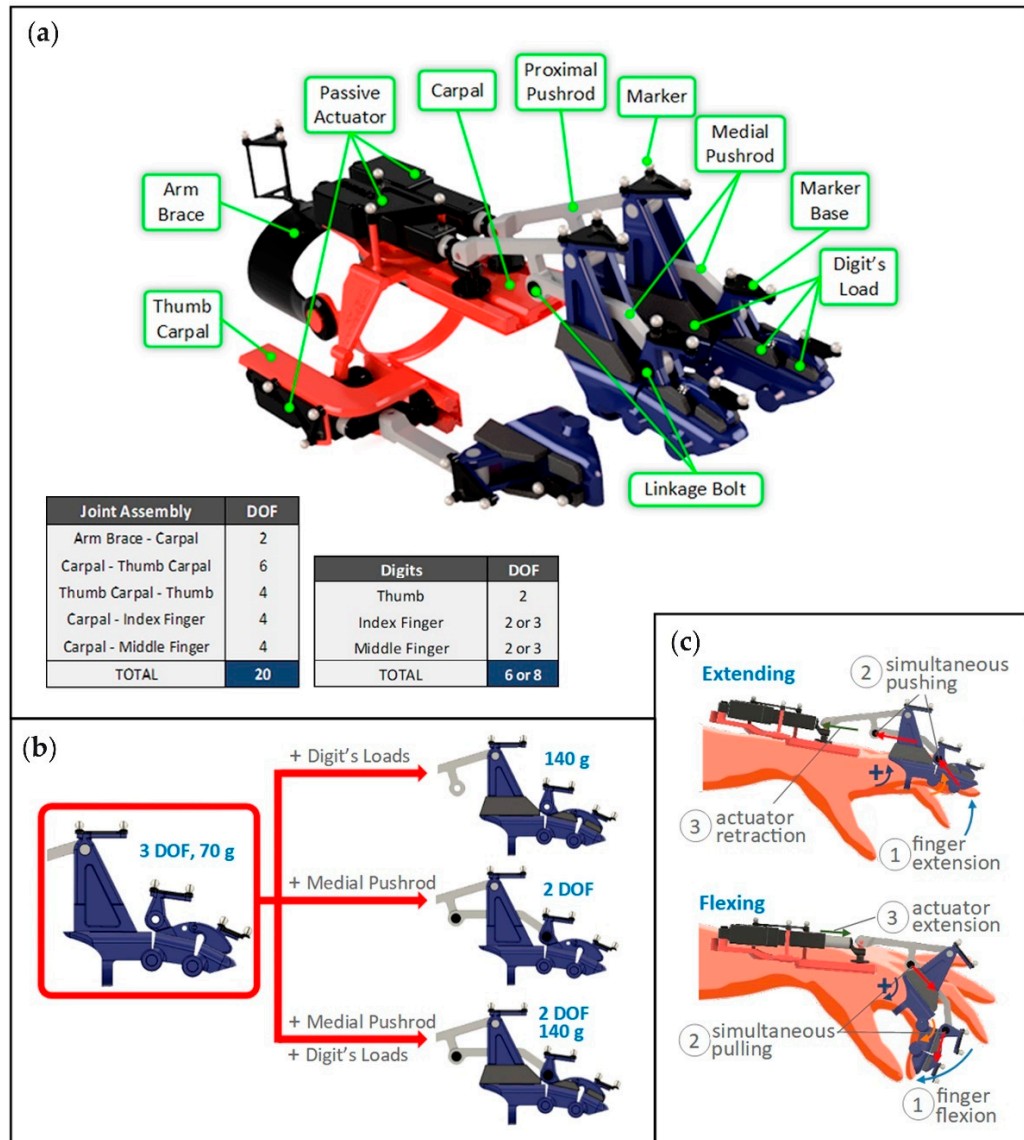

**Figure 2.** Hand exoskeleton prototype: (**a**) 3D assembly view and component's degree of freedom (DOF); (**b**) setting up digit's DOF and weight; (**c**) flexing and extending finger in 2 DOF setups.

Loads (Figure 2a,b) were attached to each segment of the exoskeleton digits to allow for weight change. The loads were made from cut tin plates and had a total weight of 70 g. The loads were made to proportionally increase the weight of the exoskeleton digits. Attaching the loads doubled the weight of the HE digits, from $70 \pm 5$ g to $140 \pm 5$ g. As shown in Figure 2b, the loads were carefully designed and placed to not increase the size of the HE digits, to allow for attachment and detachment, and not interfere with the linkage mechanism.

The prototype was designed for the participant's left hand (non-dominant hand) because the dominant hand (right hand) has higher functional abilities [28–30], which could potentially hamper the effect of the studied factors. The size of the HE prototype was customized according to the anthropometric measurements of the participant. Therefore, each participant received a HE prototype that fit properly. Additionally, all prototypes had similar motion characteristics when the two DOF modes were set by heuristic optimizations that were conducted using SAM 7.0 (Artas-Engineering, Eindhoven, The Netherlands), a

mechanism design software (Figure 3a). The finger motion characteristics were optimized to maintain the same metacarpophalangeal (MCP) joint to proximal interphalangeal (PIP) joint angle relationship among all prototypes. This relationship (Figure 3b) was adapted from a model of natural finger movement [31].

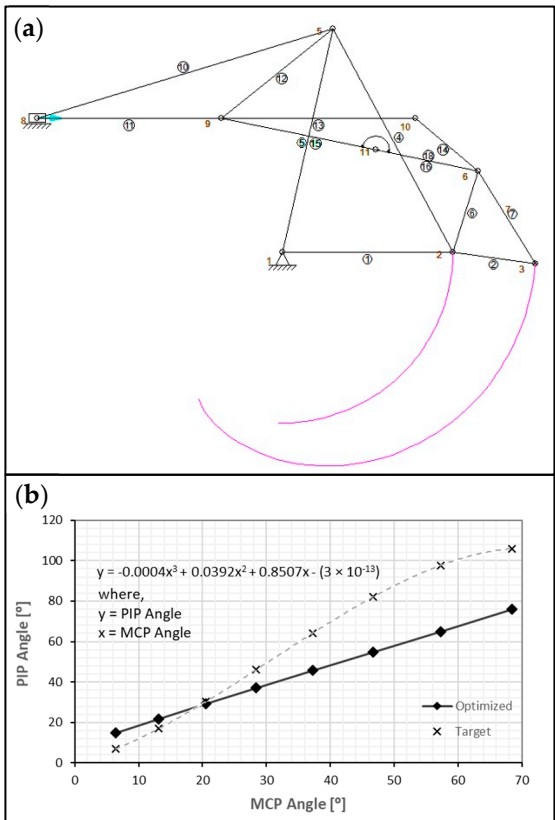

**Figure 3.** Two DOF linkage mechanism: (**a**) linkage mechanism design in SAM 7.0; (**b**) MCP to PIP joint relationship in optimized linkage versus target (natural finger movement).

For motion capture purposes, 28 retro-reflective markers with Ø4 mm (Nikon, Tokyo, Japan) were attached to the prototypes. Two or three markers for each segment were mounted on a base (a black small plate) before being attached to the prototype segments for fast installation on both the HE prototype and the barehand of the participants. The bases were made thin and light; thus, the weight addition towards the HE or hand can be neglected. When the HE is worn, the base for the markers on the proximal and medial segment (on the proximal segment for the thumb) are mounted on the top of the outer part of the linkage hinge, which is integrated with the structure of the digits (Figure 2a). This placement makes the markers move together with the digits segment without interfering with the work of the linkage mechanism. The arrangement and the list of markers on the prototype are shown in Figure 4a and those on the barehand in Figure 4b.

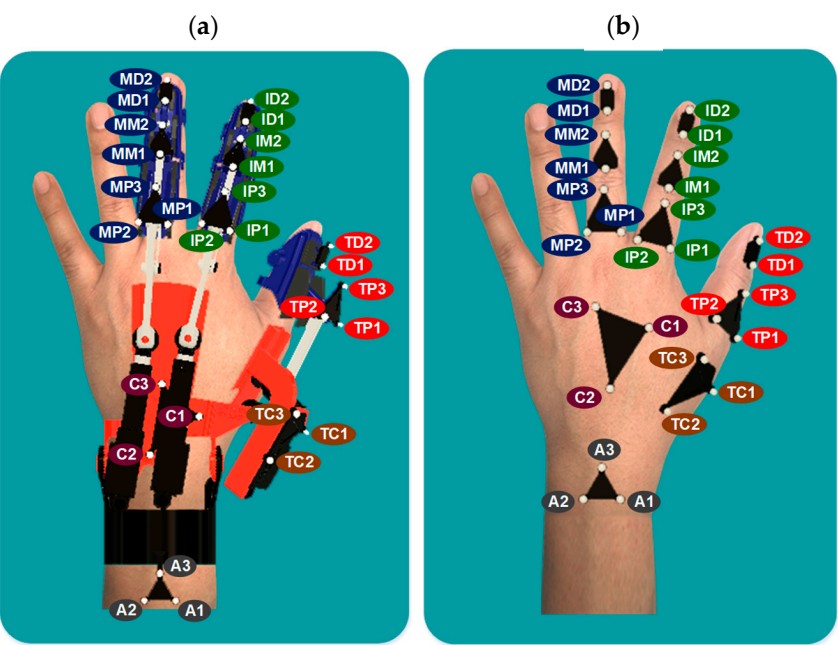

**Figure 4.** List of markers: (**a**) on hand exoskeleton; (**b**) on barehand.

### 2.3. Experiment Tasks

Two types of tasks were used in this experiment, the motion task and the productivity task. The motion task is designed to measure the range of motion (ROM) of the hand joints by analyzing the markers' position on the carpal and digits using motion analysis. Meanwhile, the productivity task is prepared to measure hand productivity through task completion time.

Pegboards for customized peg tests were prepared for the motion tasks, while a standardized nine-hole peg test (S-NHPT) [32] was selected for the productivity task. The pegboards for both tasks were prepared in two sizes, namely Ø6.4 mm peg for two-finger pinch (tip pinch) and Ø20 mm peg for three-finger pinch (tripod pinch). The criteria for choosing the Ø6.4 mm peg was based on the original nine-hole peg test [33], while that for the Ø20 mm peg was based on the proper size pinching object for the tripod pinch [34].

The customized pegboard was 500 mm long with 10 holes (Figure 5a,b). This board was designed only for a single peg to be manipulated. The peg had a flange to limit the position of the fingertip when pinching (Figure 5c). A hole was designated as the initial peg position while the nine others were the target holes. The target holes were made with three different depths, 6 mm, 12 mm, and 18 mm for target holes no. 1 to 3, no. 4 to 6, and no. 7 to 9, respectively, to stimulate the articulation of the joints of the digits. The maximum hole depths were 1 mm shallower than those of the holes on the S-NHPT to ensure that insertion could be properly performed. To prevent fatigue while reducing arm movement, an arm slider was prepared to support the participant's forearm (Figure 5d).

The S-NHPT pegboard (Figure 6a,b), consisted of two identical nine-hole peg test boards that were placed on a single wooden plate with a center-to-center distance of 18 cm. Each board was 127 × 127 mm in size. This test was designed for moving 9 pegs from one board (origin) to another board (target) in a predefined order. Figure 6c shows that the shape of the peg is cylindrical without a flange, which indicates that no specific fingertip position is required during pinching.

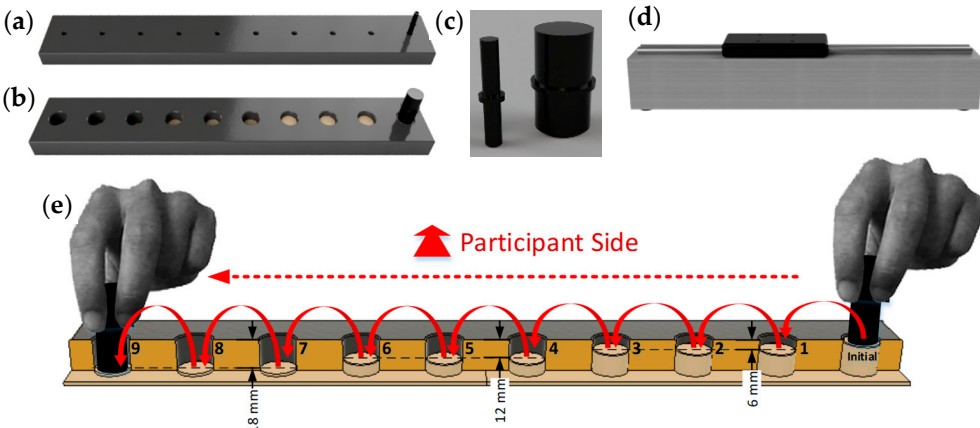

**Figure 5.** Customized pegboard test: (**a**) Ø6.4 mm pegboard for tip pinch; (**b**) Ø20mm pegboard for tripod pinch; (**c**) Ø6.4 mm and Ø20 mm peg; (**d**) the arm slider; (**e**) how to use the equipment.

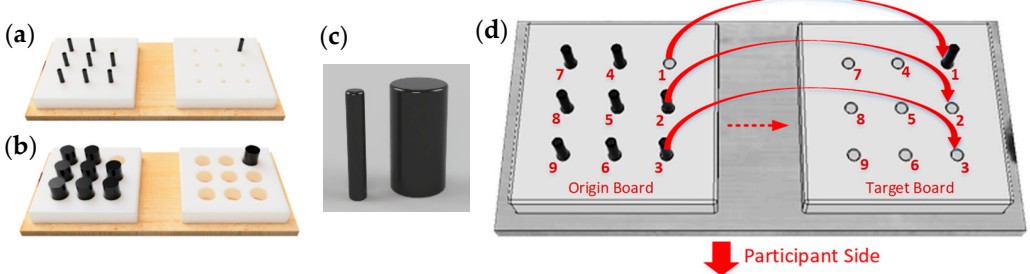

**Figure 6.** The S-NHPT: (**a**) Ø6.4 mm pegboard; (**b**) Ø20mm pegboard; (**c**) Ø6.4 mm and Ø20 mm peg; (**d**) how to use the equipment.

### 2.4. Setup and Procedure

The experiment was conducted in a dedicated space. The participant was made to sit on a chair in a comfortable upright posture while performing the task. The pegboard (both for the customized and S-NHPT pegboards) was arranged according to the reach of the participant's left hand. For the customized pegboard, the optimum position was achieved when the middle of the participant's body was in line with the position of the target hole no. 7. Meanwhile, for the S-NHPT pegboard, the optimum position was in the middle of the target board. The pegboard was placed on a 60 × 40 cm testing table while near infra-red (NiR) cameras (Motion Analysis, Rohnert Park, CA) were arranged around the table. This participant position allowed an unencumbered view of the target hole, i.e., the exoskeleton did not impair visibility. The arrangement of the experimental apparatus and equipment is presented in Figure 7a, while the participant positions are illustrated in Figure 7b,c.

Six conditions based on the HE setup (Table 1) and two conditions based on the type of pinch (the tip pinch and the tripod pinch) were used in this experiment. The execution order of the setup conditions always started and ended without the participant wearing the HE so that the wearing and taking off the HE was only done once. The four conditions for wearing the HE (3DOF NL, 3DOF L, 2DOF NL, and 2 DOF L) were balanced using the Latin Square, whereas the pinch type conditions are balanced by reversing the execution order for half of the participants.

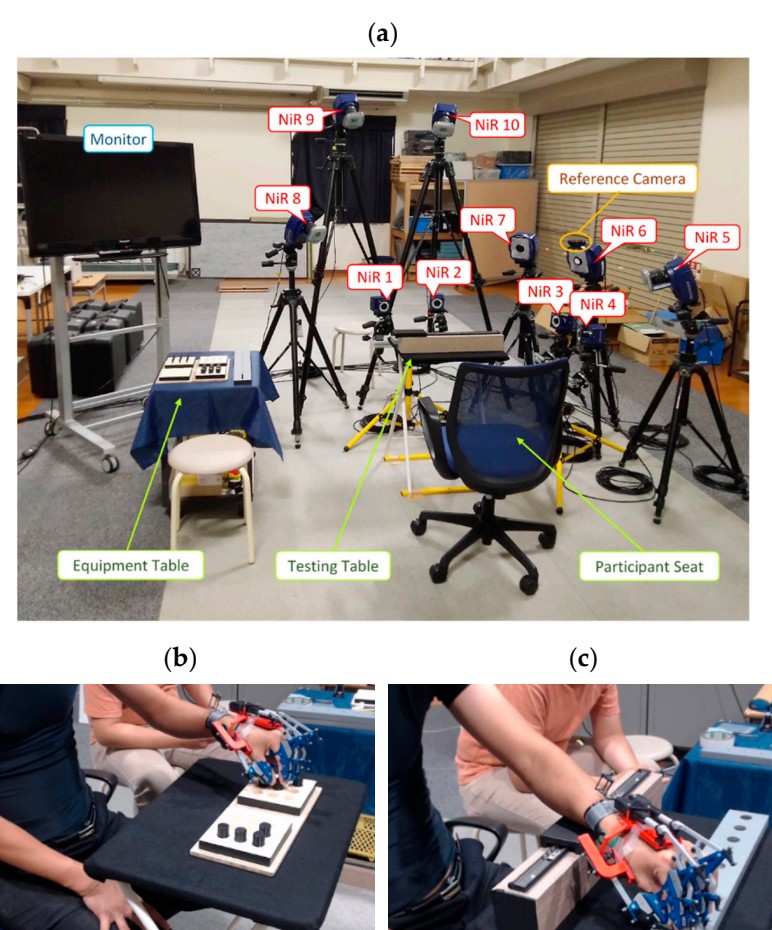

**Figure 7.** Experiment Environment: (**a**) apparatus and equipment arrangement, (**b**) participant position in productivity task, (**c**) participant position in motion task.

**Table 1.** Conditions based on hand exoskeleton setup.

| | Condition | Annotation |
|---|---|---|
| | WO1 | Condition without wearing HE (barehand) for the first run |
| balanced | 3DOF NL | Condition with wearing HE with uninstalled medial pushrod (3 DOFs linkage) and no digits' loads attached (70 g digits weight) |
| | 3DOF L | Condition with wearing HE with uninstalled medial pushrod (3 DOFs linkage) and all digits' loads attached (140 g digits weight) |
| | 2DOF NL | Condition with wearing HE with installed medial pushrod (2 DOFs linkage) and no digits' loads attached (70 g digits weight) |
| | 2DOF L | Condition with wearing HE with installed medial pushrod (2 DOFs linkage) and all digits' loads attached (140 g digits weight) |
| | WO2 | Condition without wearing HE (barehand) for the second run |

The experiment was conducted in a systematic protocol with preparation, HE set-up, pegboard set-up, practice (with instruction), the task for both pinch types, and rest. Each task was executed thrice, and the average result was calculated before the statistical analysis stage. While resting after sequentially completing the productivity and motion tasks with the same HE setup, the participants were asked to rate the perceived ease of the condition that they had just experienced (for both the tip and tripod pinches).

For the customized pegboard test, the participants were asked to insert the peg until it touched the bottom of the holes, consecutively from hole no. 1 to 9 without adjusting the grip or releasing the peg (Figure 5e). The participants were also instructed to perform the insertion motion as naturally and smoothly as possible. Conversely, in the productivity task, the participants were asked to move peg no. 1 to peg no. 9, one by one, from the origin board to the target board's corresponding holes as fast as possible. The procedure for this task is presented in Figure 6d. Because the tested hand is the left hand, the peg displacement direction was set to be opposite to the original version of the S-NHPT [32].

*2.5. Measurements*

2.5.1. Task Completion Time

A motion capture system (Motion Analysis, Rohnert Park, CA, USA) was utilized to measure the task completion times. The measurement was based on the motions of the IM2 and MM2 markers, especially along the *z*-axis. Recordings by the reference camera were utilized to verify these motions. The task completion time was calculated by subtracting the end time from the start time of a task.

2.5.2. Perceived Ease of Performing the Task

Right after the participants had a chance to experience what it was like to perform the tasks with the HE setup for both the pinch conditions (tip pinch and tripod pinch), they were asked to fill out a questionnaire about the perceived ease of performing a task. The questionnaire was rated on an 11-point scale, from 0 ("extremely hard to do") to 10 ("absolutely easy to do"), with 5 as the neutral point (neither hard nor easy). At each filling, the participant was asked to rate the perceived ease for each of the pinch types (the tip pinch and tripod pinch).

2.5.3. Task Completion Time

For measuring the digit joints' ROM, a motion capture system (Motion Analysis, Rohnert Park, CA, USA) was used. The recording rate was set to 100 Hz. The recorded marker position data were processed with the Cortex 7 software (Motion Analysis, Rohnert Park, CA, USA) to obtain continuous and clean motion data. Several virtual markers were added to allow ROM to be measured in planar motion. The names and locations of the virtual markers on the HE are presented in Figure 8a and those on the bare hand in Figure 8b. The exact positions of the virtual markers, used to produce or approximate planar motion, may differ across participants.

The data of the digit joint angles for measuring the ROM were processed using the four-marker angles method. In this method, an angle is defined by two lines that are each constructed by two markers. Table 2 below lists the digit joint angles and their forming markers.

The ROM was defined as the difference between the maximum digit joint angle and the minimum digit joint angle, which resulted from the activity of inserting a peg into a hole up to a certain depth: 6 mm, 12 mm, or 18 mm. As there are three consecutive holes with the same depth, the joint angle is measured from the moment the peg reaches the bottom of the first hole until the peg reaches the bottom of the third hole.

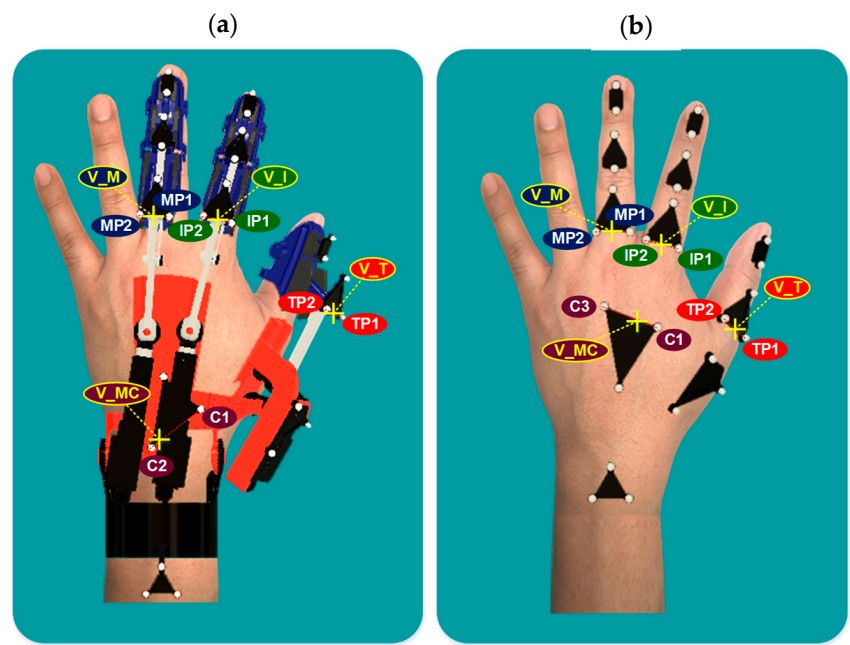

**Figure 8.** List of virtual markers: (**a**) on hand exoskeleton; (**b**) on barehand.

**Table 2.** List of digit joints and forming markers.

| Digit | Digit Joint | Four Marker Angles [Line 1, Line 2] |
|---|---|---|
| Thumb | T MCP (Thumb Metacarpophalangeal) | From V_T to TP3, From TC3 to TC2 |
| | T IP (Thumb Interphalangeal) | From TD1 to TD2, From TP3 to V_T |
| Index Finger | I MCP (Index Metacarpophalangeal) | From V_I to IP3, From V_MC to C2 (WO conditions) |
| | | From V_I to IP3, From C3 to C2 (HE conditions) |
| | I PIP (Index Proximal Interphalangeal) | From IM1 to IM2, From IP3 to V_I |
| | I DIP (Index Distal Interphalangeal) | From ID1 to ID2, From IM2 to IM1 |
| Middle Finger | M MCP (Middle Metacarpophalangeal) | From V_M to MP3, From C3 to C2 (WO conditions) |
| | | From V_M to MP3, From C3 to V_MC (HE conditions) |
| | M PIP (Middle Proximal Interphalangeal) | From MM1 to MM2, From MP3 to V_M |
| | M DIP (Middle Distal Interphalangeal) | From MD1 to MD2, From MM2 to MM1 |

*2.6. Statistical Analysis*

Prior to statistical analysis, the data of the two conditions without wearing the HE (WO1 and WO2) were averaged into a single WO condition as the baseline. For the productivity task, the baselines were used as a comparison against the conditions using the HE for the analysis of the task completion times and the perceived ease of performing the tasks, while in the motion analysis, the baselines were used to convert the ROM into ROM reduction to better show the impact of the DOF and weight factors on the ROM values. All data were checked for outliers (with Grubbs' Test) and for missing values. All statistical analyses were conducted using the SPSS 24 software (IBM Corporation, New York, NY, USA).

The measured task completion time and perceived ease of performing the task (as the dependent variable) were analyzed using the same statistical tools. Dunnett's test, at a significance level of 0.05, was utilized to analyze the difference between the conditions without wearing the HE (WO) and while wearing the HE (3DOF NL, 3DOF L, 2DOF NL,

2DOF L). Meanwhile, a two-way repeated-measures analysis of variance (ANOVA), at significance levels of 0.05 and 0.01, was applied to examine the effect of the factors (DOF and weight) on the dependent variables. To further explore the significant main effect with the significant interaction, a Bonferroni corrected pairwise comparison post-test was used. All these analyses were conducted separately according to the pinch types.

Similar to the tools used for the previous two parameters, two-way repeated-measures ANOVA was used to analyze the effect of the factors (DOF and weight) on ROM reduction, but in this case, at the 0.1 and 0.05 significance levels. A Bonferroni corrected pairwise comparison post-test was also used to further explore the significant main effect with a significant interaction. The analysis was separated based on pinch type (tip pinch, tripod pinch), depth of peg insertion (6 mm, 12 mm, 18 mm), and digit joint (T MCP, T IP, I MCP, I PIP, I DIP, M MCP, M PIP, M DIP). There were 39 separate analyses by the ANOVA.

## 3. Results

### 3.1. Task Completion Time

Based on Dunnett's test, wearing the HE showed a consistent drop in productivity with the task completion times being significantly higher in all conditions while wearing the HE compared to that at baseline. Meanwhile, the ANOVA showed that DOF reduction and weight addition significantly affect task completion time for both tip pinch and tripod pinch with no indication of interaction effects (Figure 9). The outcome of increased task completion time due to these factors was almost similar in the tripod pinch and tip pinch, with the tip pinch conditions generally requiring a higher task completion time.

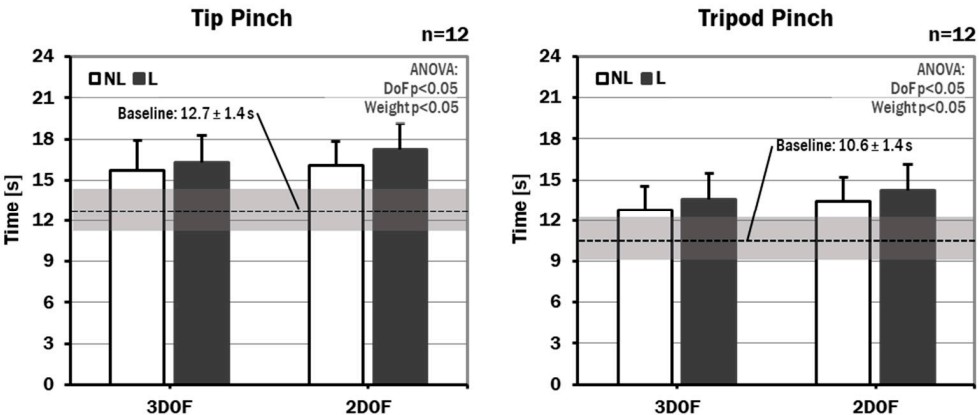

**Figure 9.** Task completion time result (lesser values are better).

### 3.2. Perceived Ease of Performing the Task

Dunnett's test indicated a significantly lower rating while wearing the HE, except for the condition with three DOF without the load (3DOF NL), compared to that at baseline. Furthermore, based on the ANOVA, both DOF and weight showed a significant effect on the rating for the perceived ease of performing the task (Figure 10). In these results, both the DOF reduction and the weight addition produced lower ratings. Meanwhile, specifically with the tripod pinch (Figure 10), it is shown that the main effects are followed by a significant interaction effect in which the rating drop due to weight addition is steeper in the 2DOF than in the 3DOF.

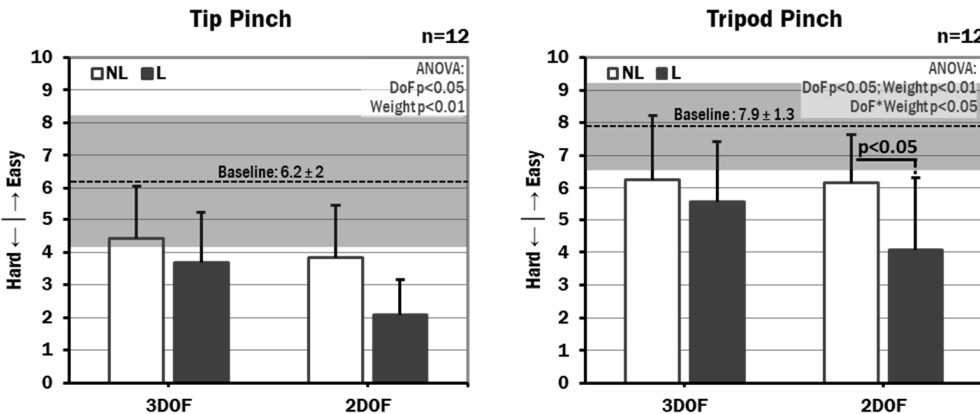

**Figure 10.** Perceived ease of performing the tasks result (higher values are better).

### 3.3. ROM Reduction of Digits' Joints

From the 39 separate analyses by ANOVA (Table 3a,b), twenty significant main effects and six interaction effects were found. These results revealed a similar pattern of factors and interaction effects between the tip and the tripod pinch. For both types of pinches, there was no significant effect on the thumb IP and the index finger DIP joint, and the significant effects were found more dominant in the finger(s) than in the thumb. However, the tripod pinch showed a greater distribution of the significant effects on the middle finger than that on the index finger.

**Table 3.** The P (T ≤ t) two-tailed value of two-way repeated measure ANOVA of ROM reduction affected by the DOF and weight (Wt.); (**a**) at tip pinch; (**b**) at tripod pinch.

| | (a) at tip pinch | | | (b) at tripod pinch | | |
|---|---|---|---|---|---|---|
| | **Insertion Depth** | | | **Insertion Depth** | | |
| | **6 mm** | **12 mm** | **18 mm** | **6 mm** | **12 mm** | **18 mm** |
| **Thumb MCP** | | | | | | |
| DOF | 0.242 | 0.096 * | 0.149 | 0.842 | 0.081 * | 0.757 |
| Wt. | 0.433 | 0.316 | 0.995 | 0.494 | 0.929 | 0.580 |
| DOF × Wt. | 0.554 | 0.985 | 0.990 | 0.695 | 0.044 ** | 0.245 |
| **Thumb IP** | | | | | | |
| DOF | 0.762 | 0.516 | 0.176 | 0.695 | 0.930 | 0.791 |
| Wt. | 0.524 | 0.124 | 0.514 | 0.251 | 0.902 | 0.474 |
| DOF × Wt. | 0.420 | 0.534 | 0.362 | 0.686 | 0.624 | 0.395 |
| **Index MCP** | | | | | | |
| DOF | 0.932 | 0.212 | 0.049 ** | 0.544 | 0.461 | 0.598 |
| Wt. | 0.996 | 0.350 | 0.912 | 0.256 | 0.133 | 0.362 |
| DOF × Wt. | 0.056 * | 0.905 | 0.457 | 0.117 | 0.211 | 0.006 ** |
| **Index PIP** | | | | | | |
| DOF | 0.001 ** | 0.005 ** | 0.001 ** | 0.011 ** | 0.071 * | 0.053 * |
| Wt. | 0.567 | 0.323 | 0.405 | 0.366 | 0.188 | 0.084 * |
| DOF × Wt. | 0.307 | 0.847 | 0.616 | 0.913 | 0.876 | 0.798 |
| **Index DIP** | | | | | | |
| DOF | 0.816 | 0.416 | 0.409 | 0.793 | 0.892 | 0.254 |
| Wt. | 0.357 | 0.104 | 0.961 | 0.652 | 0.703 | 0.506 |
| DOF × Wt. | 0.244 | 0.719 | 0.521 | 0.501 | 0.456 | 0.240 |
| **Middle MCP** | | | | | | |
| DOF | | | | 0.477 | 0.396 | 0.228 |
| Wt. | | | | 0.036 ** | 0.006 ** | 0.090 * |
| DOF × Wt. | | | | 0.319 | 0.030 ** | 0.016 ** |

**Table 3.** *Cont.*

| | (a) at tip pinch | | | (b) at tripod pinch | | |
|---|---|---|---|---|---|---|
| | Insertion Depth | | | Insertion Depth | | |
| | 6 mm | 12 mm | 18 mm | 6 mm | 12 mm | 18 mm |
| **Middle PIP** | | | | | | |
| DOF | | | | 0.002 ** | 0.016 ** | 0.006 ** |
| Wt. | | | | 0.211 | 0.051 * | 0.076 * |
| DOF × Wt. | | | | 0.268 | 0.546 | 0.191 |
| **Middle DIP** | | | | | | |
| DOF | | | | 0.501 | 0.248 | 0.112 |
| Wt. | | | | 0.647 | 0.093 * | 0.032 ** |
| DOF × Wt. | | | | 0.121 | 0.625 | 0.043 ** |

\* $p < 0.1$, \*\* $p < 0.05$.

At the tip pinch (Table 3a), no weight effect was observed in all the digit joints, and only a few signs of the DOF effect were observed in the thumb MCP joint. Meanwhile, the DOF had a significant effect on the ROM reductions of the index finger PIP joint for all the peg insertion depths, with the highest increase of 3.92 times seen at the 6 mm peg insertion depth for conditions without the load (Figure 11). At this joint, no interaction was found between DOF and weight. Further, we found that the lower the DOF, the higher the ROM reduction, as shown in Figure 11.

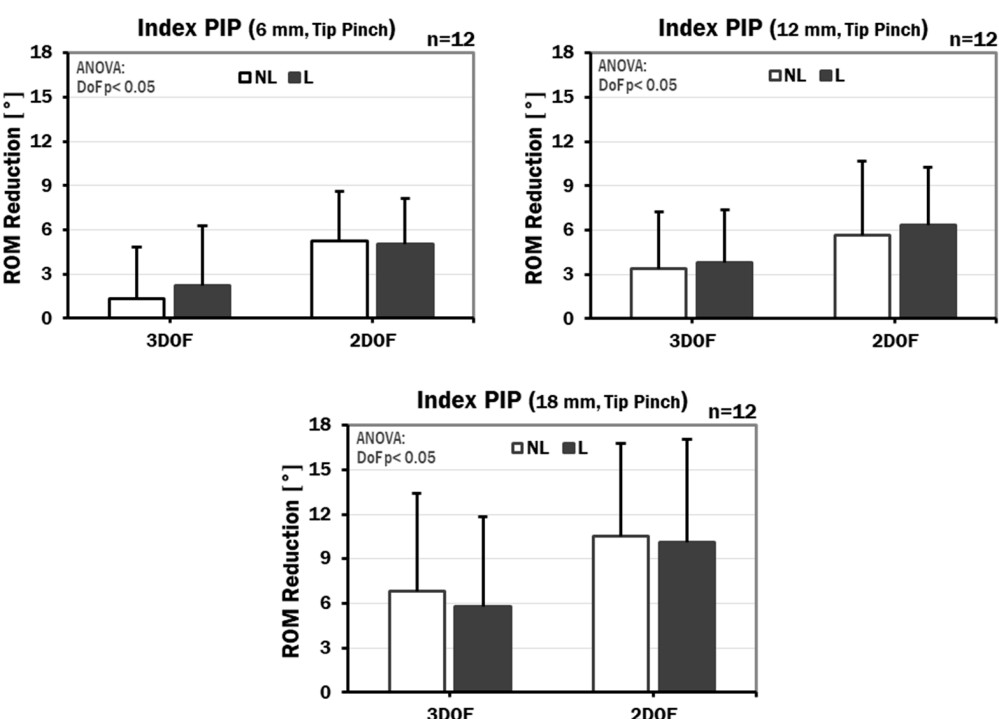

**Figure 11.** ROM reduction of the Index PIP joint.

At the tripod pinch (Table 3b), both DOF and weight showed a significant effect on more joints, and some significant main effects were even followed by significant interaction. For example, at the 12 mm peg insertion depth, at the thumb MCP joint, the DOF effect was followed by a significant interaction effect (Figure 12a) thus making the ROM reduction increase by 0.9° or 41.9% in the conditions without load. Just as the tip pinch, the DOF reduction causes a significant ROM reduction increment in the index finger PIP joint at all the peg insertion depths, without being followed by interaction effects (Table 3b). However, a more significant increment of ROM reduction up to >3° is observed in the middle finger

PIP joint at the 6 mm and 18 mm peg insertion depths (Figure 12b), while at the 12 mm insertion there was no significant difference even though it had a similar trend.

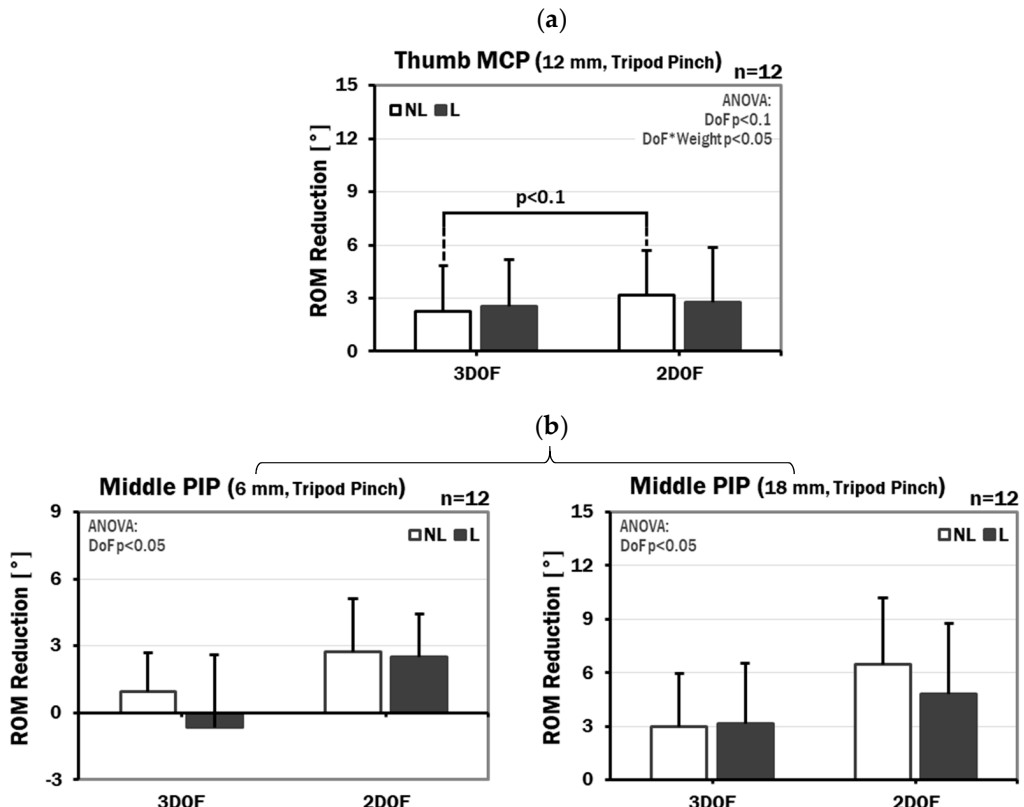

**Figure 12.** ROM reduction of joints affected by the degree of freedom reduction for insertion using tripod pinch; (**a**) at the thumb MCP joint; (**b**) at the middle finger PIP joint.

In contrast to the DOF reduction effect, the weight addition effect in the tripod pinch (Figure 13), commonly produced ROM reduction correction. Moreover, the effect of weight addition was more significant in the two DOF conditions. In Figure 12a, a 15.7% correction of ROM reduction ($p < 0.1$) due to weight addition is found after further exploration of the significant interaction effect at the index finger MCP joint. Significant weight addition effects followed by significant interaction effects were also indicated at the middle finger MCP joint. At this joint, ROM reductions were lower at 2 DOF compared to 3 DOF with correction by up to 0.9° or 29% at 12 mm peg insertion (Figure 13b). At the middle finger DIP joint, the correction was even greater, up to 3.8°, thus making the ROM at the same level as the baseline (Figure 13c).

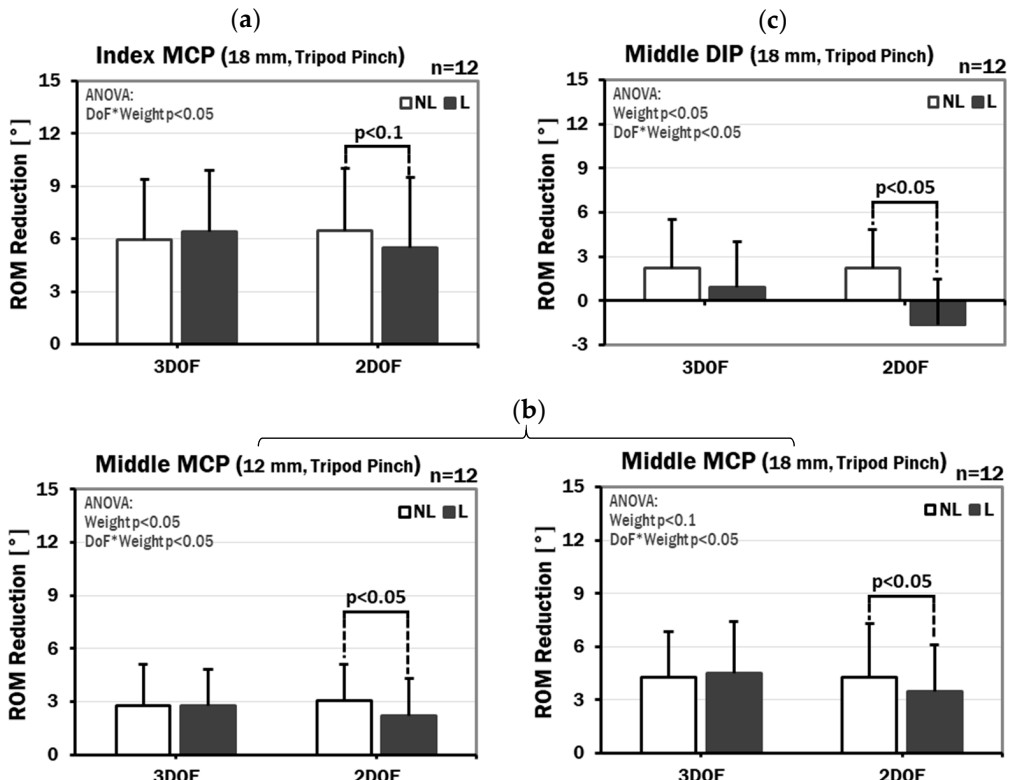

**Figure 13.** ROM reduction of joints affected by weight addition for insertion using tripod pinch; (**a**) at the index finger MCP joint; (**b**) at the middle finger MCP joints; (**c**) at the middle finger DIP joint.

## 4. Discussion

This study aims to investigate the effects of the HE DOF and weight on the hand joint mobility function in performing fine hand use activities that were measured through task completion time, perceived ease of performing the task, and the ROM. The changes in productivity, perceived ease, and ROM that have been found need to be further discussed.

### 4.1. Effect of Wearing HE

Both productivity and perceived ease of performing the task were significantly affected by wearing the HE. With this result, it can be stated that wearing the HE reduced one aspect of joint mobility function, namely the ease of movement. The unavoidable consequences of wearing an HE, such as movement restriction or overall hand weight increase, were suspected to be the cause of this reduction in performance. This result is in line with that of other studies regarding the effect of movement restriction on hand [17–19]. However, based on the subjective evaluation results (Figure 8), wearing an HE while performing the tasks was not perceived as difficult, as long as the task was not too precise (performed with the tripod pinch), and the participant did not wear the 2DOF L setup (heavy and involving high resistance). These findings are in agreement with a study of a rigid type HE, with which the user expressed feelings of ease [35]. This result showed that the HE has the potential to become more comfortable to wear while performing certain fine activities by adjusting some design parameters, even though in general, wearing the HE would reduce productivity.

### 4.2. Effect of DOF

Due to the pushrods in the exoskeleton, the segments are restricted to move in tandem with one another for 2DOF and 3DOF, as shown in Figure 3. Our results demonstrated that the degree of restriction significantly affects the ease of movement for both the tip and the tripod pinch (Figures 9 and 10). This is consistent with the results of a previous study [18]

where the application of splints to the hand joints can reduce productivity significantly. However, in Table 3, it is shown that the effect did not occur at every joint. The DOF effects were significant only at the PIP joint for all the insertion depths.

The 2DOF setup used in this study imposed potential movement restrictions due to the medial push rod. Eventually, the finger ROM is reduced as shown in Figures 11 and 12. The PIP joint naturally has higher mobility compared to the MCP and DIP joints when performing gripping action [31], making it easily disturbed by external factors that inhibit motion. Figure 11 is consistent with this DOF effect since the two DOF linkage systems used in this study regulated the motion of the PIP joint more than the other joints. Predictably, the DIP joint was the only joint that was not affected by DOF reduction, since its ROM is the smallest.

On the other hand, the impact of DOF reduction on insertion tasks could be accomplished with opposable thumb presents, the abduction/adduction of fingers could be maintained, as well as wrist flexion/extension movements [36]. Meanwhile, movement stability during a decrease in finger movement was achieved via the thumb, as their opposed movement partner (Table 3a,b and Figure 12a). However, the effect was less noticeable since the ROM of the thumb joints is smaller than that of the fingers and it did not affect the success of performing the task.

In the end, it can be stated that first, the negative effect of the DOF reduction occurred in any condition, with any speed, and any level of precision; second, the decrease in joint mobility function due to the DOF reduction would be most imposed on joints that had a large ROM and were directly regulated by the linkage system, and third, there was no noticeable compensation on the other digits' joints due to PIP ROM reduction, and this did not interfere with task completion.

### 4.3. Effect of Weight

Figure 9 shows that weight addition significantly lengthens the task completion time. This could be due to the increase in HE weight causing a higher demand for movement control from the hand-arm muscles, especially when high acceleration or deceleration is required [37]. This simulates the decrease in hand movement performance when a heavy glove is worn [19]. From a mechanics point of view, the slowed hand performance occurs because the increased hand inertia tends to resist changes in motion direction which ultimately makes it difficult to control. Difficulty in controlling the movement will interfere with the mobility function, eventually reducing hand dexterity.

Besides productivity, the rating of perceived ease of performing the task was also lowered by weight addition (Figure 10). The effect was even stronger than DOF reduction with a $p < 0.01$ significant level. The possible cause of this result was that the subjective evaluation was aimed at not only the productivity task but also the motion tasks. Therefore, it is related to the longer task duration of the use of the HE, in which the weight addition psychophysically affected the capacity limit of the user [38], especially if the user had not gotten used to it.

From the result of the dexterity test and subjective evaluation, it can be said that the HE's weight affects hand joint mobility function in terms of its ease of movement. Additionally, the psychophysical effect of weight may occur when the heavy HE is worn for a long duration during early use.

### 4.4. Interaction Effect

Significant interaction effects between the DOF and weight were found at tripod pinch. In Figure 10, the effect of weight works differently between 3DOF and 2DOF. This figure shows that a massive rating drop occurs when the 2DOF setup is applied. At this pinch type, not only is the rating significantly different but it also jumps down from easy to difficult (from 6 to 4). The possible cause of this occurrence is that the preserved flexibility in 3 DOF might have been allowing the hand to perform better and be perceived as much easier to move, even though the weight had been increased. The preserved flexibility

was also described as the cause of better performance in a proposed protective gloves design [39].

A critical finding of the interaction effect was presented in the motion analysis results. In Table 3, the interaction occurs at the MCP and DIP joints and has never been indicated at the PIP joint. As it has been studied [31], the MCP and DIP joints have relatively lower ROM than PIP joints during gripping and manipulation. This means that the interaction is more likely to occur at the lower ROM joints. Moreover, this interaction frequently follows the significant weight effect, and the effect is positive for ROM (Figure 13). Interestingly, the significant difference due to these effects mostly occurs at the two DOF. A possible explanation is that the linkage resistance of the 2DOF setup is greater than 3DOF, where the assistance against the motion resistance upon an increase in weight becomes more meaningful.

The possible cause of the recovered ROM due to weight addition at 2 DOF is the gravitational advantage of the downward peg insertion. This assumption is based on the proof that aligning the direction of force with gravity can be used as a method to improve the ROM [40], and it has even been used as a principle in manual working [41]. The force produced by the load's mass and gravity successfully assisted the peg insertion motion despite the high movement resistance of 2 DOF linkages. Lastly, comprehensive assistance is important for peg insertion because it requires both flexion and extension. In this case, this force successfully accommodated both, meaning it was great enough to assist the insertion (downward) and small enough to not resist the motion when the direction was reversed (upward).

At this point, it can be stated that the weight addition at 2 DOF brought a positive effect to the movement range of the joint mobility function. Gravity and appropriate weight addition play an important role in making this happen. However, digit weight has become an increasingly important factor in this setup, and it requires careful consideration in its management.

*4.5. Design Direction*

Lightweight and high DOF are the most ideal design characteristics of an effective HE for daily life activities. However, a strong and simple device with these characteristics is difficult to design due to technical and technological reasons. Therefore, based on our findings, we formulated a compromise strategy to design a more effective HE.

Choosing between 3- and 2-DOF fingers requires careful consideration because each has its advantages and disadvantages. When a high-performance HE system is required or high power is important, 2 DOF is recommended because it can distribute the power to two segments with a single actuator. We could try to reduce the negative effect of this choice (movement resistance) by using a flexible material [42] for improving the ROM of the PIP joints or unfixing some digit joints [43]; however, that might come at the expense of decreased mechanical stability [44].

Considering the role of gravity in two DOF, it seems beneficial for a HE to provide a small force for counterbalancing movement resistance in addition to its main force for gripping assistance. The implementation of this idea needs a creative process such as "separation" as a solution to the physical contradiction that might occur [45]. In this case, separation can mean the use of two different types of actuators for two different purposes or two different control systems with different sensors sensitivity. However, it should be noted that these methods have the potential to increase the design complexity of the HE.

*4.6. Limitation and Future Study*

This study has several limitations. In general, the HE aims to improve hand function, regardless of age, sex, or disability. However, in this study, we only recruited adult male participants with normal hand function. On the other hand, fine hand use activities in daily life involve various types of objects when performing various tasks. However, the tasks in

this study were still only for handling the peg (simple cylindrical object) and performing the pegboard tests with only two levels in each factor included.

It has been found that the ROM is improved due to a force that can counterbalance the movement resistance. However, in this study, the force is the consequence of the digits' weights that were assisted by the direction of gravity. This advantage can be expanded by providing a counterbalancing force from an external source such as active actuators. Hence, we suggest future studies that use a power-assisted HE prototype with an adjustable actuation force. In the case of measuring the various object manipulation task, motion measurement methods other than camera-based (not affected by the object blocking), such as the use of the IMU (inertial measuring unit) module [46,47] are a potential option.

## 5. Conclusions

In general, wearing an HE reduces the mobility function especially when a lower DOF setup is applied. However, the weight addition may improve the movement range aspect of the function. Additionally, managing the digit's weight becomes increasingly important when the low DOF concept is considered. Considering the movement restriction generated by the HE mechanical system, the counterbalancing force might be a potential solution, and a further study of its characteristics is needed.

This study emphasizes the basic needs of an HE as a wearable assistive device that is light and flexible (has a high DOF). However, when heavy systems and low DOF are unavoidable, overcoming motion barriers becomes an important requirement. Based on the findings in this study, the usage of a counterbalancing force that works either passively or actively becomes a recommendation. This is a differentiator from the general HE design strategy which tends to apply advanced materials or design concepts based on unfixed joints.

**Author Contributions:** Conceptualization, I.P., W.L.Y., P.Y.L. and S.M.; methodology, I.P.; software, I.P.; validation, W.L.Y. and S.M.; formal analysis, I.P.; investigation, I.P.; resources, I.P.; data curation, I.P.; writing—original draft preparation, I.P.; writing—review and editing, W.L.Y., P.Y.L. and S.M.; visualization, I.P.; supervision, S.M.; project administration, S.M.; funding acquisition, S.M. All authors have read and agreed to the published version of the manuscript.

**Funding:** This work was supported by the Japan Society for the Promotion of Science (JSPS) KAK-ENHI Grant Numbers JP17H01454 and JP21H04898.

**Data Availability Statement:** Not applicable.

**Acknowledgments:** We thank Hiroki Nakashima, Shin Takesue, and Jeewon Choi (Kyushu University) for providing assistance in the motion capture system setup. We thank Sachi Takahashi (Kyushu University) for the help in the tools and materials procurement.

**Conflicts of Interest:** The authors declare no conflict of interest.

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
