# Peer review of "The Effect of the Degree of Freedom and Weight of the Hand Exoskeleton on Joint Mobility Function"

_robotics, doi:10.3390/robotics11020053_

Round 1

Reviewer 1 Report

The paper presents the effect of the degree of freedom and weight of the hand exoskeleton on the Joint Mobility Function hand. The paper is well written.

  1. On rows 50-52 the authors say they have simply identified two influencing factors “two mechanical design characteristics that have the potential to influence fine hand activities while wearing an HE” not indicating which studies were the basis for this conclusion
  1. Further explanations are required for the relation between the kinematic of the exoskeleton and the ROM analysis system. How was it implemented and supported the analysis process?
  1. Please briefly detail also in the conclusion your finding.  What does it add to the subject area compared with other published material?
  2. After these minor revisions regarding this crucial point, the paper is ready for publication in my opinion.

Reviewer 2 Report

In this paper, the authors investigated the effects of the degree of freedom (DOF) and weight of the hand exoskeleton (HE) on hand joint mobility function by using experiments. The results from the experiments surely can prove some valuable things for such a hand exoskeleton.

However, this paper looks like an experiment report, not a research paper.

As a reviewer, I think some theoretical analysis should be added to this paper and compared with the experiment results.

Otherwise, this paper should be written as short communication for such a hand exoskeleton.

Reviewer 3 Report

Thank you very much for sending the article titled: The Effect of the Degree Of Freedom and Weight of the Hand Exoskeleton on Joint Mobility Function. I honestly admit that it is rare to find such a well-written article. Generally, the paper is very interesting to my mind, however, the authors should refer to the following statements:

  • please write the last paragraph of the Introduction focused on the article scheme.
  • at the beginning of the Materials and Methods  section please include a Flowchart
  • authors used in the experiment NiR and Reference cameras. Could you think about using Inertial measurement units in your prototype? They are small, cheap and authors receive parameters (like Euler angles). For example, in article titled:  Dynamic Model of a Humanoid Exoskeleton of a Lower Limb with Hydraulic Actuators, Sensors 2021, authors used IMU for obtain these parameters. To improve quality of this article (although I believe it is well written) you can cite above article and write a few sentences about it in the Discussion section as a future research. 

Round 2

Reviewer 1 Report

Accept in present form

Reviewer 3 Report

Thank you very much for improving your manuscript.